# Convergence Rates for the Constrained Sampling via Langevin Monte Carlo

**DOI:** 10.3390/e25081234

**Published:** 2023-08-18

**Authors:** Yuanzheng Zhu

**Affiliations:** School of Statistics, Southwestern University of Finance and Economics, Chengdu 611130, China; zhuyz0626@smail.swufe.edu.cn

**Keywords:** Bayesian computation, constrained sampling, convex support, Langevin Monte Carlo, MCMC, mixing time bound

## Abstract

Sampling from constrained distributions has posed significant challenges in terms of algorithmic design and non-asymptotic analysis, which are frequently encountered in statistical and machine-learning models. In this study, we propose three sampling algorithms based on Langevin Monte Carlo with the Metropolis–Hastings steps to handle the distribution constrained within some convex body. We present a rigorous analysis of the corresponding Markov chains and derive non-asymptotic upper bounds on the convergence rates of these algorithms in total variation distance. Our results demonstrate that the sampling algorithm, enhanced with the Metropolis–Hastings steps, offers an effective solution for tackling some constrained sampling problems. The numerical experiments are conducted to compare our methods with several competing algorithms without the Metropolis–Hastings steps, and the results further support our theoretical findings.

## 1. Introduction

Sampling from distributions with some constraints has extensive applications in statistics, machine-learning, and operations research, among other areas. Some distributions have bounded support, such as the simple but versatile uniform distribution which serves as the foundation for a series of Monte Carlo methods, as discussed in [1]. Furthermore, many statistical inference problems involve estimating parameters subject to constraints on the parameter space, which defines a posterior distribution with bounded support in a Bayesian setting. Examples include Latent Dirichlet Allocation [2], truncated data problems in failure and survival time studies [3], ordinal data models [4], constrained lasso and ridge regressions [5], and non-negative matrix factorization [6]. In Bayesian learning, sampling from posterior distributions is a fundamental primitive, used for exploring posterior distributions, identifying the unknown parameters [7,8], obtaining credible intervals, and solving inverse problems [7,8]. Finally, constrained sampling has great potential in solving constrained optimization problems [9,10].

Many Markov Chain Monte Carlo (MCMC) algorithms have been extensively studied for sampling from probability distributions with convex support or more generally with constrained parameters, mainly in the fields of Bayesian statistics and theoretical computer science. Early work includes, among others, [1,11,12,13,14]. Firstly, based on MCMC algorithms, a direct solution involves discarding samples that violate the constraints, thereby exclusively retaining samples that satisfy the constraints; see, for example, [1,15,16]. However, these rejection-type approaches may encounter an excessive number of rejections or an extremely large acceptance rate within some local subspace that satisfies the constraints, which leads to poor mixing and computational inefficiency, especially for complicated constraints and the high dimensional distributions [17,18]. Secondly, some literature draws inspiration from penalty functions in optimization problems and considers the construction of barriers along the boundaries of the constrained domain, effectively constraining the sampling process within the constrained area. These approaches encounter a major challenge when the samples reach the boundaries of the constraints, necessitating the implementation of a mechanism based on reflection to redirect them back into the constrained region. To address this issue, Ref. [19] extended the Hamiltonian Monte Carlo (HMC) method by setting the potential energy outside the constraint region to infinity, restricting the states to the desired domain. Ref. [20] extended the HMC method to sample from truncated multivariate Gaussian distributions, and Ref. [21] proposed an approach that involves mapping the constrained domain onto a sphere in an augmented space. Thirdly, motivated by the constrained optimization methods, the constrained sampling problem can be reformulated as an unconstrained sampling problem via suitable transformations. Following this idea, Ref. [22] proposed a family of novel algorithms based on HMC through the introduction of Lagrange multipliers that address a broader range of constrained sampling problems. More recently, Ref. [23] tackled the constrained sampling problem via the mirror-Langevin algorithm. In spite of the widespread adoption of these MCMC methods, most of them have primarily focused on the algorithm design and lack the rigorous theoretical analysis of convergence rates.

Among all the MCMC algorithms, a class of algorithms based on the Langevin dynamics has garnered significant attention in both practical applications and theoretical analyses [24,25,26,27]. It has recently witnessed a notable increase in non-asymptotic analyses of these algorithms, initiated by the seminal work of [28]. In the setting of unconstrained sampling, Ref. [29] extended the theoretical analysis of convergence rates by studying with decreasing step size, and Refs. [30,31] derived corresponding convergence results based on alternative distances. These theoretical analyses focus on the Langevin algorithm without the Metropolis–Hastings step. More recently, Refs. [32,33] have shown that incorporating the Metropolis–Hastings step can significantly improve the convergence rate of the associated Langevin algorithm. In the setting of constrained sampling, Ref. [34] suggested a Euclidean projection step in the Langevin algorithms for the constrained case (PLMC) and derived the convergence rate of the associated Markov chain. Ref. [35] presented a detailed theoretical analysis for a proximal version of the Langevin algorithm that incorporates the Moreau-Yosida envelope of the indicator function (MYULA) to handle the distributions that are restricted to a convex body. Ref. [36] constructed the mirrored Langevin algorithm (MLD) using a mirror map to constrain the domain, which achieves the same convergence rate as its unconstrained counterpart [28]. However, these constrained sampling algorithms are all developed based on the Langevin algorithm without incorporating the Metropolis–Hastings steps, thus not leveraging the fast mixing advantages of them.

In this paper, we considered the constrained Langevin Monte Carlo with the Metropolis–Hastings step for sampling from the distributions restricted to some convex support. Firstly, for certain constraints, we re-examine the simple and intuitive rejection-type methods for sampling from constrained distributions, and reach a surprising discovery that the corresponding algorithm still retained the advantage of rapid convergence by carefully selecting the step size parameter. Subsequently, for the more generally constrained domain, we build upon the framework proposed in [35], incorporating the Metropolis–Hastings step for further refinement, and analyze the convergence rate of the corresponding Markov chain. We present detailed non-asymptotic analysis for these constrained algorithms and achieve notably enhanced convergence rates in the total variation distance. Compared with the best rate in [36], our results show that adopting the Metropolis–Hastings step in some constrained MCMC algorithms can also lead to an exponentially improved dependence on the error tolerance.

The rest of the paper is organized as follows. In Section 2, we introduce the preliminaries and the problem set-up of our study. Then, we propose the constrained sampling algorithms tailored to different types of constraint regions in Section 3. Section 4 provides the non-asymptotic theoretical results of the proposed algorithms. The numerical experiments and comparisons are presented in Section 5. Some Markov chain basics are provided in Appendix A and all the technical proofs are deferred to Appendix B.

Notation: Let ⌈a⌉ represent the smallest integer not less than a∈R. For a vector $x\in R^{d}$, we use ${\left|x\right|}_{2}$ to denote its Euclidean norm. For a q×q symmetric matrix *A*, denote by $\lambda_{\min}\left(A\right)$ and $\lambda_{\max}\left(A\right)$ the smallest and largest eigenvalues of *A*, respectively, and let $A^{T}$ be its transpose. For two square matrices *A* and *B*, we write A⪯B if (B−A) is a positive semi-definite matrix. Denote by I(·) the indicator function. For r>0, let $B\left(x,r\right)=\{y\in R^{d}:|y-x{|}_{2}\leq r\}$ denote a closed Euclidean ball with center *x* and radius *r*. For two real-valued sequences $a_{n}$ and $b_{n}$, we say $a_{n}=O\left(b_{n}\right)$ if there exists a universal constant *c* such that $a_{n}\leq cb_{n}$, and $a_{n}=\tilde{O}\left(b_{n}\right)$ if $a_{n}\leq c_{n}b_{n}$ where the sequence $c_{n}$ grows at most poly-logarithmically with *n*. For any two probability measures μ and ν, denote by ${∥\mu -\nu ∥}_{\mathrm{TV}}$ the total variation distance between μ and ν.

## 2. Preliminaries and Problem Set-Up

In this section, we introduce the MCMC sampling methods with its mixing analysis, the traditional unconstrained Metropolis-Adjusted Langevin Algorithm (MALA), and our problem set-up for this paper.

### 2.1. Markov Chain Monte Carlo and Mixing

Consider a distribution Π equipped with a density $\pi :R^{d}\mapsto R_{+}$ such that
(1)$$\pi \left(x\right)\propto e^{-U(x)}$$
for some potential function $U:R^{d}\mapsto R$. In certain scenarios, it is necessary to perform sampling from this distribution. For example, many statistical applications involve estimating the expectation of a function g(X) for X∼π, where analytical and numerical computation is infeasible. Monte Carlo approximation provides a solution by generating samples from Π and using sample mean to estimate the population expectation. Hence, the key point is to access samples from Π.

MCMC represents a class of popular sampling algorithms, which construct an appropriate Markov chain whose stationary distribution is Π or close to Π in certain metrics. The class of the Metropolis–Hastings algorithms refers to a type of MCMC method that ensures the corresponding Markov chain converges to the target distribution by incorporating the Metropolis–Hastings step. The Metropolis–Hastings algorithms usually take two steps to generate a Markov chain: a proposal step and a reject-accept step. At each iteration, a sample is generated from the proposal distribution in the proposal step, and it is updated as a new state of the Markov chain with probability determined by the Metropolis–Hastings correction in the reject-accept step.

Given an error tolerance ε∈(0,1), in order to obtain an ε-accurate sample with respect to some metric, one simulates the Markov chain for a certain number of steps *k*, as determined by a mixing time analysis. Specifically, we are concerned about how many steps the chain needs to take such that the current distribution of the chain is ε-close to the target distribution Π. Based on this, we define the ε-mixing time with respect to the target distribution Π as
(2)$$\tau \left(\varepsilon ;P^{0},\Pi \right)=\min\{k\in N:∥T^{k}\left(P^{0}\right)-\Pi {∥}_{\mathrm{TV}}\leq \varepsilon \}$$
for the error tolerance ε∈(0,1), where T is the transition operator of the Markov chain and $T^{k}\left(P^{0}\right)$ is the distribution of the Markov chain at *k*-th step from an initial distribution $P^{0}$.

### 2.2. Metropolis-Adjusted Langevin Algorithm

Consider the problem of sampling from the distribution with density defined as (Equation 1). MALA [26,27] adopts the Gaussian distribution $N\{x_{k}-h\nabla U\left(x_{k}\right),2hI_{p}\}$ as the proposal distribution for the *k*-th step, where $x_{k}$ is the current state and h>0 is a proper step size, and performs a Metropolis–Hastings accept-reject step. MALA is the standard Metropolis–Hastings algorithm applied to the Langevin dynamics, and the associated Langevin-type algorithms belong to a family of gradient-based MCMC sampling algorithms [37]. The Langevin-type algorithms can be understood as the Euler discretization of the Langevin dynamics:$$dX_{t}=-\nabla U\left(X_{t}\right)\;dt+\sqrt{2}\;dW_{t}\;,$$
where $W_{t}\left(t\geq 0\right)$ is the standard Brownian motion on $R^{d}$.

Algorithm 1 provides the unconstrained MALA for sampling from the distribution supported on $R^{d}$, where $\phi_{h}\left(\cdot \;|\;x\right)$ denotes the probability density function of $N\{x-h\nabla U\left(x\right),2hI_{d}\}$.
**Algorithm 1** Metropolis-adjusted Langevin algorithm**Input:** a sample $x^{0}\in R^{d}$ from an initial distribution $P^{0}$, the step size *h*  **for** 
k=0,1,2,…,K−1
 **do**      **Proposal step:** $y^{k+1}\leftarrow x^{k}-h\nabla U\left(x^{k}\right)+\xi$, where $\xi ∼N(0,2hI_{p})$      **Accept-reject step:**      compute $\alpha^{k+1}=\min\left\{1,\frac{\phi_{h}\left(x^{k}\;|\;y^{k+1}\right)\pi \left(y^{k+1}\right)}{\phi_{h}\left(y^{k+1}\;|\;x^{k}\right)\pi \left(x^{k}\right)}\right\}$      sample $u^{k+1}$ from the uniform distribution on [0,1]      **if** $\alpha^{k+1}\geq u^{k+1}$, **then** $x^{k+1}\leftarrow y^{k+1}$      **else**$\;\;x^{k+1}\leftarrow x^{k}$      **end if**  **end for****Output:** 
$x^{1},x^{2},\ldots ,x^{K}$

### 2.3. Problem Set-Up

In this part, we consider the problem of sampling from a target distribution or posterior $\Pi^{*}$ supported on a compact set $X\subset R^{d}$ equipped with a density $\pi^{*}$. It can be written in the form
(3)$$\pi^{*}\left(x\right)=\frac{\exp\{-U(x)\}I(x\in X)}{\int_{X}\exp\left\{-U\left(y\right)\right\}\;dy}$$
for some potential function $U:R^{d}\mapsto R$. Assume that the function U(·) and the set X satisfy the following assumptions:

**Assumption** **1.**
*U(·) is a twice continuously differentiable, L-smooth and m-strongly convex function on $R^{d}$. That is, there exist universal constants L≥m>0 such that*

$$\frac{m}{2}{\left|y-x\right|}_{2}^{2}\leq U\left(y\right)-U\left(x\right)-{\left\{\nabla U\left(x\right)\right\}}^{T}\left(y-x\right)\leq \frac{L}{2}{\left|y-x\right|}_{2}^{2}$$

*for any $x,y\in R^{d}$.*


**Assumption** **2.**
*$X\subset R^{d}$ is a compact and convex set satisfying*

$$B\left(x^{*},r\right)\subset X\subset B\left(x^{*},R\right)$$

*for some universal constants 0<r≤R and $x^{*}\in X$.*


Hereafter, we assume that the above two assumptions hold, which is frequently used in the literature for the analysis of constrained sampling algorithms [34,35,36]. We will modify the MALA in Algorithm 1 to adapt to sampling from the above constrained distribution, and analyse its non-asymptotic theoretical properties and derive the mixing time bound in terms of the problem dimension *d* and the error tolerance ε.

## 3. The Constrained Langevin Algorithms

In this section, we present three sampling algorithms based on MALA to handle the distribution constrained within some convex body X. As discussed in [34], the inherent challenges in constrained sampling problems arise from the complex properties on the boundary of the constraint region, and the lack of the curvature in the potential function. To tackle these challenges, Ref. [34] initially studied constrained sampling from the uniform distribution on X, and then extended the exploration to more general distributions. Similarly, we begin our investigation by examining some simple constrained regions and progressively extend our analysis to more complex constraint scenarios.

### 3.1. Constrained Langevin Algorithm via Rejection

We initially discuss the case where the constraint region X is an Euclidean ball on $R^{d}$, where the boundary can be characterized by a curve equation. If $X=B(x^{*},R)$ for some universal constant R>0 and $x^{*}\in R^{d}$, we consider the simple and intuitive rejection-type methods via the Metropolis–Hastings accept-reject step for sampling from the distribution with density defined as (Equation 3). The constrained MALA for $X=B(x^{*},R)$ outlined in Algorithm 2 as follows, where $\phi_{h}\left(\cdot \;|\;x\right)$ denotes the probability density function of the Gaussian distribution $N\{x-h\nabla U\left(x\right),2hI_{d}\}$.
**Algorithm 2** The MALA for Euclidean ball constrained domain**Input:** a sample $x^{0}\in X$ from an initial distribution $P^{0}$, the step size *h*  **for**  
k=0,1,2,…,K−1 
**do**      **Proposal step:** $y^{k+1}\leftarrow x^{k}-h\nabla U\left(x^{k}\right)+\xi$, where $\xi ∼N(0,2hI_{p})$      **Accept-reject step:**      **if** $y^{k+1}\in X$ **then**          compute $\alpha^{k+1}=\min\left\{1,\frac{\phi_{h}\left(x^{k}\;|\;y^{k+1}\right)\pi^{*}\left(y^{k+1}\right)}{\phi_{h}\left(y^{k+1}\;|\;x^{k}\right)\pi^{*}\left(x^{k}\right)}\right\}$          sample $u^{k+1}$ from the uniform distribution on [0,1]          **if** $\alpha^{k+1}\geq u^{k+1}$, **then** $x^{k+1}\leftarrow y^{k+1}$          **else**$\;\;x^{k+1}\leftarrow x^{k}$          **end if**      **else**$\;\;x^{k+1}\leftarrow x^{k}$      **end if**  **end for****Output:** 
$x^{1},x^{2},\ldots ,x^{K}$

Compared with Algorithm 1, this modified algorithm forces the Markov chain to stay at the current state when it jumps out of the limited state space $X=B(x^{*},R)$, which is a quite natural extension of the unconstrained MALA. This idea is not completely novel. Ref. [34] suggested a projection step in unadjusted Langevin algorithm for sampling from a log-concave distribution with compact support. Ref. [10] proposed an MALA for constrained optimization, where they used a similar step to constrain the Markov chain to stay at a given state space. Due to the favorable properties on the boundary of constrained domain $X=B(x^{*},R)$, we can establish the theoretical results of Algorithm 2; see Lemma A1 in Appendix B for details.

### 3.2. Norm-Constrained Domain

Regularization is a technique commonly used in machine-learning and statistical modeling. As discussed in [38], some models with regularization can be reformulated as the distributions with norm-constraint on the parameters. Notice that the $L_{p}$-norm for the vector $x={\left(x_{1},x_{2},\ldots ,x_{d}\right)}^{T}\in R^{d}$ is defined as
$${\left|x\right|}_{p}=\left\{\begin{array}{cccc} \; & {\left(\sum_{i=1}^{d}{\left|x_{i}\right|}^{p}\right)}^{1/p}\;, & \; & p\in (0,\infty )\\ \; & \max_{1\leq i\leq d}\left|x_{i}\right|\;, & \; & p=\infty \;. \end{array}\right.$$
For the norm-constrained domain $X=\{x\in R^{d}:|x{|}_{p}\leq C\}$ with some universal constant C>0, we can transform it into the Euclidean ball B(0,1) via a vector-valued function f:X↦B(0,1). Specifically, for any $x={\left(x_{1},x_{2},\ldots ,x_{d}\right)}^{T}\in X$, we have $y=f\left(x\right)=:{\left\{f_{1}\left(x\right),f_{2}\left(x\right),\ldots ,f_{d}\left(x\right)\right\}}^{T}$ with
$$f_{i}\left(x\right)=\left\{\begin{array}{cccc} \; & C^{-p/2}\mathrm{sgn}\left(x_{i}\right){\left|x_{i}\right|}^{p/2}\;, & \; & p\in (0,\infty )\\ \; & x_{i}\frac{{\left|x\right|}_{\infty }}{{C|x|}_{2}}\;, & \; & p=\infty \end{array}\;,\;\;\;\;\;\;1\leq i\leq d\right.$$
such that y∈B(0,1). Due to the bijective nature of the function f:X↦B(0,1), its inverse function $f^{-1}=:g:B\left(0,1\right)\mapsto X$ can be defined accordingly. Similarly, for any $y={\left(y_{1},y_{2},\ldots ,y_{d}\right)}^{T}\in B\left(0,1\right)$, we have $x=g\left(y\right)=:{\left\{g_{1}\left(y\right),g_{2}\left(y\right),\ldots ,g_{d}\left(y\right)\right\}}^{T}$ with
$$g_{i}\left(y\right)=\left\{\begin{array}{cccc} \; & C\mathrm{sgn}\left(y_{i}\right){\left|y_{i}\right|}^{2/p}\;, & \; & p\in (0,\infty )\\ \; & Cy_{i}\frac{{\left|y\right|}_{2}}{{\left|y\right|}_{\infty }}\;, & \; & p=\infty \end{array}\;,\;\;\;\;\;\;1\leq i\leq d\right.$$
such that x∈X. By utilizing the vector-valued functions f(·) and g(·) defined above, we can employ the Euclidean ball constrained sampling algorithm, as described in Section 3.1, to tackle the norm-constrained domain $X=\{x\in R^{d}:|x{|}_{p}\leq C\}$. The computational process is outlined in Algorithm 3, where
$$\pi^{B(0,1)}\left(x\right)=\frac{\exp\{-U(x)\}I\{x\in B(0,1)\}}{\int_{B(0,1)}\exp\left\{-U\left(y\right)\right\}\;dy}$$
with the potential function U(·).
**Algorithm 3** The MALA for norm-constrained domain**Input:** a sample $x^{0}\in X$ from an initial distribution $P^{0}$, the step size *h*  **for** 
k=0,1,2,…,K−1 
**do**      **Transformation step:** $y^{k}\leftarrow f\left(x^{k}\right)$      **Proposal step:** $z^{k+1}\leftarrow y^{k}-h\nabla U\left(y^{k}\right)+\xi$, where $\xi ∼N(0,2hI_{p})$      **Accept-reject step:**      **if** $z^{k+1}\in B\left(0,1\right)$ **then**          compute $\alpha^{k+1}=\min\left\{1,\frac{\phi_{h}\left(y^{k}\;|\;z^{k+1}\right)\pi^{B(0,1)}\left(z^{k+1}\right)}{\phi_{h}\left(z^{k+1}\;|\;y^{k}\right)\pi^{B(0,1)}\left(y^{k}\right)}\right\}$          sample $u^{k+1}$ from the uniform distribution on [0,1]          **if** $\alpha^{k+1}\geq u^{k+1}$, **then** $y^{k+1}\leftarrow z^{k+1}$          **else**$\;\;y^{k+1}\leftarrow y^{k}$          **end if**      **else**$\;\;y^{k+1}\leftarrow y^{k}$      **end if**      **Transformation step:** $x^{k+1}\leftarrow g\left(y^{k+1}\right)$  **end for****Output:** 
$x^{1},x^{2},\ldots ,x^{K}$

Compared with Algorithm 2, the Algorithm 3 achieves the X→B(0,1)→X transformation by incorporating two transformation steps, thereby addressing the norm-constrained sampling problems. The main purpose of this approach is to facilitate theoretical analysis by leveraging the well-understood properties of the boundary of the Euclidean ball compared to the boundary of the norm-constrained domain; see Section B.7 for details.

### 3.3. Constrained Langevin Algorithm via an Approximation of the Indicator Function

We proceed to discuss the constrained sampling for more general constraint regions. Given $X\in R^{d}$, define
(4)$$\iota_{X}\left(x\right)=:-\log\left\{I\left(x\in X\right)\right\}=\left\{\begin{array}{cccc} \; & 0\;, & \; & \mathrm{If}\;x\in X\\ \; & \infty \;, & \; & \mathrm{If}\;x\notin X \end{array}\right.$$
for any $x\in R^{d}$. Then, the target distribution $\Pi^{*}$ with density defined as (Equation 3) can be reformulated as
(5)$$\pi^{*}\left(x\right)=\frac{\exp\{-V_{X}\left(x\right)\}}{\int_{X}\exp\left\{-V\left(y\right)\right\}\;dy}$$
with the potential function $V_{X}:R^{d}\mapsto R$ satisfying
(6)$$V_{X}\left(\cdot \right)=U\left(\cdot \right)+\iota_{X}\left(\cdot \right)\;,$$
where $\iota_{X}\left(\cdot \right)$ is defined in (Equation 4). Notice that $\iota_{X}\left(\cdot \right)$ is a convex function on $R^{d}$. Under Assumption 1, we then know that the potential function $V_{X}\left(\cdot \right)$ is smooth and strongly convex on $R^{d}$. By this transformation, the problem of constrained sampling is apparently converted into an unconstrained counterpart. However, the non-differentiability of the function $V_{X}\left(\cdot \right)$ on the boundary of X poses a challenge when applying the gradient-based unconstrained sampling algorithms. To address this issue, we can approximate the function $\iota_{X}\left(\cdot \right)$ by a differentiable function such as the Moreau-Yosida (MY) envelope [35]. The MY envelope of $\iota_{X}\left(\cdot \right)$ is defined as
(7)$$\iota_{X}^{\lambda }\left(x\right)=\underset{y\in R^{d}}{\inf}\{\iota_{X}\left(x\right)+{\left(2\lambda \right)}^{-1}{\left|x-y\right|}_{2}^{2}{\}=\left(2\lambda \right)}^{-1}{\left|x-{\mathrm{Pro}}_{X}\left(x\right)\right|}_{2}^{2}$$
for any $x\in R^{d}$, where λ>0 is a regularization parameter and ${\mathrm{Pro}}_{X}\left(\cdot \right)$ is the projection function onto X. By [35], the function $\iota_{X}^{\lambda }\left(\cdot \right)$ is convex and continuously differentiable with the gradient
(8)$$\nabla \iota_{X}^{\lambda }\left(x\right)=\lambda^{-1}\left\{x-{\mathrm{Pro}}_{X}\left(x\right)\right\}$$
for any $x\in R^{d}$, and it holds that
(9)$$|\nabla \iota_{X}^{\lambda }\left(x\right)-\nabla \iota_{X}^{\lambda }{\left(y\right)|}_{2}\leq \lambda^{-1}{\left|x-y\right|}_{2}$$
for any $x,y\in R^{d}$. Then the approximation of $V_{X}\left(\cdot \right)$ defined as (Equation 6) can be given by
(10)$$V_{X}^{\lambda }\left(\cdot \right)=U\left(\cdot \right)+\iota_{X}^{\lambda }\left(\cdot \right)\;,$$
which is continuously differentiable, smooth and strongly convex on $R^{d}$ if U(·) satisfying Assumption 1. Define the distribution $\Pi^{*,\lambda }$ with density
(11)$$\pi^{*,\lambda }\left(x\right)=\frac{\exp\{-V_{X}^{\lambda }\left(x\right)\}}{\int_{R^{d}}\exp\left\{-V^{\lambda }\left(y\right)\right\}\;dy}\;.$$
Recall that the target distribution $\Pi^{*}$ with the reformulated density defined as (Equation 5). As discussed in [35], under some mild conditions including Assumptions 1 and 2, the approximation error between $\Pi^{*}$ and $\Pi^{*,\lambda }$ in total variation distance can be made arbitrarily small by adjusting the regularization parameter λ. Therefore, we can utilize the gradient-based unconstrained sampling algorithms, such as the MALA presented in Algorithm 1, for constructing an appropriate Markov chain whose stationary distribution is close to $\Pi^{*}$; see Algorithm 4 for details, where $\phi_{h}^{\lambda }\left(\cdot \;|\;x\right)$ denotes probability density function of the Gaussian distribution $N\{x-h\left\{\nabla U\left(x\right)+\nabla \iota_{X}^{\lambda }\left(x\right)\right\},2hI_{d}\}$ with $\nabla \iota_{X}^{\lambda }\left(\cdot \right)$ defined as (Equation 8).
**Algorithm 4** The MALA for convex constrained domain**Input:** a sample $x^{0}\in R^{d}$ from an initial distribution $P^{0}$, the step size *h*  **for** 
k=0,1,2,…,K−1 
**do**      **Proposal step:** $y^{k+1}\leftarrow x^{k}-h\left\{\nabla U\left(x^{k}\right)+\nabla \iota_{X}^{\lambda }\left(x^{k}\right)\right\}+\xi$, where $\xi ∼N(0,2hI_{p})$      **Accept-reject step:**      compute $\alpha^{k+1}=\min\left\{1,\frac{\phi_{h}^{\lambda }\left(x^{k}\;|\;y^{k+1}\right)\pi^{*,\lambda }\left(y^{k+1}\right)}{\phi_{h}^{\lambda }\left(y^{k+1}\;|\;x^{k}\right)\pi^{*,\lambda }\left(x^{k}\right)}\right\}$      sample $u^{k+1}$ from the uniform distribution on [0,1]      **if** $\alpha^{k+1}\geq u^{k+1}$, **then** $x^{k+1}\leftarrow y^{k+1}$      **else**$\;\;x^{k+1}\leftarrow x^{k}$      **end if**  **end for****Output:** 
$x^{1},x^{2},\ldots ,x^{K}$

## 4. Theoretical Results

In this section, we first analyze the properties of the Markov chains determined by the three constrained sampling algorithms presented in Section 3, and then establish the mixing time bounds of these Markov chains.

### 4.1. Properties of the Markov Chains

The outcomes $\left\{x^{1},\ldots ,x^{K}\right\}$ from each algorithm presented in Section 3 form a Markov chain, whose properties are established in Propositions 1, 2, and 3, respectively, as below.

**Proposition** **1.***For $X=B(x^{*},R)$ with some universal constant R>0 and $x^{*}\in R^{d}$, the Markov chain determined by Algorithm* 2 *is $\Pi^{*}$-irreducible, smooth, and reversible with respect to the stationary distribution $\Pi^{*}$ with density $\pi^{*}$ defined as* (Equation 3) *(The definition of the $\Pi^{*}$-irreducible, reversible, and smooth Markov chain is deferred to*
Appendix A*).*

**Remark** **1.***Proposition* 1 *shows that the Markov chain determined by Algorithm* 2 *enjoys a series of nice properties as the unconstrained MALA, which form the basis for the study of the mixing time bounds of such Markov chain.*

The similar properties hold for the Markov chains determined by Algorithms 3 and 4 as well.

**Proposition** **2.***For $X=\{x\in R^{d}:|x{|}_{p}\leq C\}$ with some universal constant C>0, the Markov chain determined by Algorithm* 3 *is $\Pi^{*}$-irreducible, smooth, and reversible with respect to the stationary distribution $\Pi^{*}$ with density $\pi^{*}$ defined as* (Equation 3).

**Proposition** **3.***Under Assumption* 2,* the Markov chain determined by Algorithm* 4 *is $\Pi^{*,\lambda }$-irreducible, smooth, and reversible with respect to the distribution $\Pi^{*,\lambda }$ with density $\pi^{*,\lambda }$ defined as* (Equation 11).

### 4.2. Mixing Time Bounds of the Markov Chains

For a distribution Π supported on $X\subset R^{d}$ with the density π, recall that the ε-mixing time with respect to Π is defined as (Equation 2). A β-warm initial distribution $P^{0}$ with density $p^{0}$ with respect to the distribution Π is commonly used for the mixing time analysis, which satisfies
$$\underset{x\in X}{\sup}\frac{p^{0}\left(x\right)}{\pi (x)}\leq \beta$$
for some finite constant β>0. We say that the Markov chain is ς-lazy if at each iteration the chain is forced to stay at the previous state with probability at least ς. It is a convenient assumption for theoretical analysis of the convergence rate, but not likely to be used in practice since the lazy steps slow down the mixing rate of Markov chain. Given the definitions above and some Markov chain basics in Appendix A, we can obtain the following results for some well-behaved Markov chains defined on {X,B(X)}.

**Lemma** **1.***Consider a reversible, *Π*-irreducible, ς-lazy, and smooth Markov chain defined on {X,B(X)} with stationary distribution *Π* supported on X. For any error tolerance ε∈(0,1) and β-warm initial distribution $P^{0}$, the ε-mixing time with respect to *Π* satisfying*$$\tau \left(\varepsilon ;P^{0},\Pi \right)\leq \left\lceil \frac{4}{\varsigma }\int_{4\beta^{-1}}^{\varepsilon^{-2}}\frac{\;dv}{v{\tilde{\Omega }}^{2}\left(v\right)}\right\rceil \;,$$*where $\tau (\varepsilon ;P^{0},\Pi )$ and $\tilde{\Omega }\left(\cdot \right)$ are defined, respectively, in* (Equation 2) *and* (Equation 15).

**Remark** **2.***Lemma* 1 *provides a control on the mixing time of a Markov chain on X in terms of $\tilde{\Omega }\left(\cdot \right)$. This result can be seen as an extension of Lemma 3 in* [33] *to the case where a Markov chain defined on {X,B(X)}. We then can readily derive the mixing time bound if a lower bound for $\tilde{\Omega }\left(\cdot \right)$ is known.*

The following lemma gives a lower bound for Ω(·).

**Lemma** **2.***Assume that the distribution *Π* supported on X with the density π satisfy the log-isoperimetry inequality defined as* (Equation 12) *for some constant $\hat{c}>0$. If a reversible Markov chain with stationary distribution *Π* satisfies $\sup_{{x,y\in X:\;|x-y|}_{2}\leq \Delta }{∥T_{x}-T_{y}∥}_{\mathrm{TV}}\leq 1-\delta$ for some δ∈(0,1) and Δ>0, it then holds that*
$$\Omega \left(v\right)\geq \frac{\delta }{4}\min\left\{1,\;\frac{\Delta }{4\hat{c}}\log^{1/2}\left(1+\frac{1}{v}\right)\right\}$$
*for any v∈(0,1/2], where $T_{x}$ is the one-step transition distribution of this Markov chain at x∈X and Ω(·) is the conductance profile of this Markov chain defined in* (Equation 14).

**Remark** **3.***Lemma* 2 *states a lower bound for the conductance profile of a Markov chain on X. Similar results can be found in the* [33,39,40]. *Lemma* 2, *together with Lemma* 1, *provides a general framework for obtaining mixing time bound of a well-behaved Markov chain on X*.

Based on Lemmas 1 and 2, we can drive the upper bounds for each ε-mixing time of the Markov chains determined by the three constrained sampling algorithms presented in Section 3.

**Theorem** **1.***For $X=B(x^{*},R)$ with some universal constant R>0 and $x^{*}\in R^{d}$, let Assumption* 1 *hold with $L^{3/8}R^{3/4}\geq 16/\sqrt{d}+8$ and $L^{-15/8}m^{2}R^{1/4}\geq 12d$. Given a β-warm initial distribution $P^{0}$ and an error tolerance ε∈(0,1), the Markov chain determined by Algorithm* 2 *satisfies*
$$\tau \left(\varepsilon ;P^{0},\Pi^{*}\right)=O\left(\frac{L^{7/4}R^{3/2}d}{m}\log\frac{\log\beta }{\varepsilon }\right)$$
*for any step size h satisfying*
$$\frac{1}{L^{7/4}R^{3/2}d}\leq h\leq \min\left[\frac{R^{2}{\left(1-\tilde{c}\right)}^{2}}{4{\left\{\log^{1/2}\left(16/u\right)+\sqrt{d}\right\}}^{2}},\;\frac{\sqrt{u}}{4\sqrt{3}L^{3/2}R},\;\frac{u}{128L{\left\{\log^{1/2}\left(16/u\right)+\sqrt{d}\right\}}^{2}}\right]$$
*with $\tilde{c}={\left\{1+\left(L^{-7/2}R^{-3}d^{-2}-L^{-11/4}R^{-3/2}d^{-1}\right)m^{2}\right\}}^{1/2}$ and some constant u∈(1/2,1), where $\Pi^{*}$ with density $\pi^{*}$ defined as* (Equation 3).

**Remark** **4.***Theorem* 1 *presents a sharp mixing time bound for Algorithm* 2 *with a β-warm initial distribution as $\tilde{O}\left\{d\log\left(1/\varepsilon \right)\right\}$ up to β and L, m, R which are specified in Assumptions* 1 *and* 2. *This result improves upon the previously known mixing time bounds for constrained sampling algorithms in* [34,35,36]; *see*
Table 1
*for details*.

For sampling from the norm-constrained domain $X=\{x\in R^{d}:|x{|}_{p}\leq C\}$ with some universal constant C>0, we transform it into the sampling from Euclidean ball B(0,1) as shown in Algorithm 3; then, the similar result holds for the Markov chain determined by Algorithm 3 as well.

**Corollary** **1.***For $X=\{x\in R^{d}:|x{|}_{p}\leq C\}$ with some universal constant C>0, let Assumption* 1 *hold with $L^{3/8}\geq 16/\sqrt{d}+8$ and $L^{-15/8}m^{2}\geq 12d$. Given a β-warm initial distribution $P^{0}$ and an error tolerance ε∈(0,1), the Markov chain determined by Algorithm* 3 *satisfies*
$$\tau \left(\varepsilon ;P^{0},\Pi^{*}\right)=O\left(\frac{L^{7/4}d}{m}\log\frac{\log\beta }{\varepsilon }\right)$$
*for any step size h satisfying*
$$\frac{1}{L^{7/4}d}\leq h\leq \min\left[\frac{{\left(1-\bar{c}\right)}^{2}}{4{\left\{\log^{1/2}\left(16/u\right)+\sqrt{d}\right\}}^{2}},\;\frac{\sqrt{u}}{4\sqrt{3}L^{3/2}},\;\frac{u}{128L{\left\{\log^{1/2}\left(16/u\right)+\sqrt{d}\right\}}^{2}}\right]$$
*with $\bar{c}={\left\{1+\left(L^{-7/2}d^{-2}-L^{-11/4}d^{-1}\right)m^{2}\right\}}^{1/2}$ and some constant u∈(1/2,1), where $\Pi^{*}$ with density $\pi^{*}$ defined as* (Equation 3).

For the Markov chain determined by Algorithm 4, we can also derive a sharp mixing time bound by the mixing time analysis for sampling from log-concave distribution without constraints in [33] and the approximation error between $\Pi^{*}$ and $\Pi^{*,\lambda }$ in [35].

**Theorem** **2.***Let Assumptions* 1 *and* 2 *hold, and assume that there exists a universal constant $\tilde{C}>0$ such that $\exp\left\{\inf_{x\in X^{c}}U\left(x\right)-\sup_{x\in X}U\left(x\right)\right\}\geq \tilde{C}$. Given the initial distribution $P^{0}=N\left\{x^{★},{\left(L+\lambda^{★-1}\right)}^{-1}I_{d}\right\}$ with $x^{★}=\arg\min_{x\in R^{d}}V_{X}^{\lambda^{★}}\left(x\right)$ and an error tolerance ε∈(0,1), the Markov chain determined by Algorithm* 4 *satisfies*
$$\tau \left(\varepsilon ;P^{0},\Pi^{*}\right)=O\left[\frac{(L+\lambda^{★-1})d}{m}\log\frac{d}{\varepsilon }\cdot \max\left\{1,\sqrt{\frac{L+\lambda^{★-1}}{dm}}\right\}\right]$$
*for the step size h satisfying*
$$h=c\frac{1}{\left(L+\lambda^{★-1}\right)d\cdot \max\left\{1,\sqrt{\frac{L+\lambda^{★-1}}{dm}}\right\}}$$
*with some universal constant c>0, where $V_{X}^{\lambda^{★}}\left(\cdot \right)$ is defined as in *(Equation 10)* with $\lambda^{★}:=8\pi^{-1}\varepsilon^{2}r^{2}d^{-2}{\tilde{C}}^{2}$, and $\Pi^{*}$ with density $\pi^{*}$ defined as* (Equation 3).

**Remark** **5.***Theorem* 2 *presents a mixing time bound for Algorithm* 4 *with a feasible initial distribution as $O\{d^{3}\varepsilon^{-2}\log\left(d/\varepsilon \right)\}$ up to L, m, r which are specified in Assumptions *1* and *2* if we choose the regularization parameter $\lambda =\lambda^{★}$. This result improves upon the mixing time bound for constrained sampling algorithm without incorporating the Metropolis–Hastings step in* [35]; *see*
Table 1
*for details*.

## 5. Numerical Experiments

In this section, we conduct numerical experiments to validate the theoretical properties derived in Section 4 and compare the constrained sampling algorithms presented in Section 3 with three competing MCMC algorithms for sampling from constrained log-concave distributions listed in Table 1 under various simulation settings. The implementation of these algorithms involves the selection of a step size. For Algorithms 2 and 3, we follow Theorem 1 and Corollary 3, respectively, to select the step size. For Algorithm 4, we choose the step size as that in [32] for the MALA for sampling from log-concave distribution without constraints. The step size choice of the other three MCMC algorithms follows the recommendation in the associated papers; see Table 2 for details.

### 5.1. Sampling from the Euclidean Ball Constrained Domain

We consider the problem of sampling from a truncated multivariate Gaussian distribution on X, which admits the density
$$\pi^{*}\left(x\right)\propto \exp\left\{-\frac{{\left(x-\mu \right)}^{T}\Sigma^{-1}\left(x-\mu \right)}{2}\right\}I\left(x\in X\right)\;,$$
where the mean μ=0 and covariance matrix $\Sigma \in R^{d\times d}$ is a diagonal matrix with $\lambda_{\max}\left(\Sigma \right)=10$ and $\lambda_{\min}\left(\Sigma \right)=1$. For this target distribution, the potential function U(·) and its derivatives are given as $U\left(x\right)=2^{-1}x^{T}\Sigma^{-1}x$, $\nabla U\left(x\right)=\Sigma^{-1}x$, and $\nabla^{2}U\left(x\right)=\Sigma^{-1}$. Therefore, the function U(·) is smooth with parameter $L=\lambda_{\min}^{-1}\left(\Sigma \right)$ and strongly convex with parameter $m=\lambda_{\max}^{-1}\left(\Sigma \right)$ on $R^{d}$. We select X=B(0,R) with R=5, the initial distribution $P^{0}=N_{X}\left\{0,{\left(2L\right)}^{-1}I_{d}\right\}$, and use the inverse transformation algorithm [14] to generate an initial point from $P^{0}$. We compare Algorithm 2 with the three sampling algorithms in literature given in Table 2, and follow the recommendation in the associated papers to choose the initial points of the three sampling algorithms.

#### 5.1.1. The Trace Graphs of Sampling Algorithms

To initiate a preliminary assessment of the convergence properties of these algorithms, we commence with simple sample trace plots. Write $x={\left(x_{1},\ldots ,x_{d}\right)}^{T}\in R^{d}$ and $\mu ={\left(\mu_{1},\ldots ,\mu_{d}\right)}^{T}\in R^{d}$. Figure 1 depicts the traces of $x_{1}$ of the Markov chains determined by the four sampling algorithms under dimension d=10. Evidently, in comparison to the other three algorithms, Algorithm 2 exhibits a notably faster mixing time, as evidenced by the trace consistently remaining around its mean $\mu_{1}=0$. Conversely, the traces of the other three sampling algorithms exhibit greater fluctuations and deviate more from $\mu_{1}=0$.

Figure 2 illustrates the histograms and densities corresponding to these traces of $x_{1}$. Similarly, it is evident that Algorithm 2 achieves sample means closer to $\mu_{1}=0$, along with the least variance. Conversely, the sample means obtained from the other three sampling algorithms exhibit a certain degree of deviation from $\mu_{1}=0$, accompanied by heavier tails.

#### 5.1.2. Dimension and Error Dependence of Algorithm 2

The goal of this simulation is to demonstrate that the dimension and error tolerance dependence of the mixing time bound for Algorithm 2 both conform to the theoretical results shown in Theorem 1.

Since the total variation distance between continuous measures is hard to estimate, we use the error in quantiles along some direction for convergence diagnostics in the experiments. In the spirit of [33], we measure the error in the 95% quantile of the sample distribution and the true distribution in the direction along the eigenvector of Σ corresponding to $\lambda_{\min}\left(\Sigma \right)$. The approximate mixing time ${\hat{k}}_{\mathrm{mix}}\left(\varepsilon \right)$ is then defined as the smallest iteration *k* when such error between the distribution of the Markov chain at iteration *k* and the target distribution falls below the error tolerance ε. We simulate 20 independent runs of the Markov chain of the algorithms with *N* = 20,000 samples at each run to determine the approximate mixing time ${\hat{k}}_{\mathrm{mix}}\left(\varepsilon \right)$. Then the final ${\hat{k}}_{\mathrm{mix}}\left(\varepsilon \right)$ is the average of these 20 independent runs.

Figure 3a shows the dependence of the approximate mixing time ${\hat{k}}_{\mathrm{mix}}\left(0.2\right)$ as a function of dimension *d* for Algorithm 2. By the linear regression for ${\hat{k}}_{\mathrm{mix}}\left(0.2\right)$ with respect to *d*, we conclude that the mixing time of Algorithm 2 is linear in *d* with slope 4.137 and *R*-squared 0.991. Figure 3b presents the dependence of the approximate mixing time ${\hat{k}}_{\mathrm{mix}}\left(\varepsilon \right)$ on the inverse of the error tolerance $\varepsilon^{-1}$ for Algorithm 2 under d=4. The linear regression for the approximate mixing time ${\hat{k}}_{\mathrm{mix}}\left(\varepsilon \right)$ with respect to $\varepsilon^{-1}$ suggests that the mixing time of Algorithm 2 is linear in $\log(\varepsilon^{-1})$ with slope 15.854 and *R*-squared 0.994, which is consistent with the theoretical results given in Theorem 1.

#### 5.1.3. Comparison with Competitive Algorithms

Figure 4a shows the dependence of the approximate mixing time ${\hat{k}}_{\mathrm{mix}}\left(0.2\right)$ on the problem dimension *d* for the four sampling algorithms. Compared with the other three algorithms, the approximate mixing time of Algorithm 2 seems more robust to dimension. When *d* is small, the approximate mixing time of the four algorithms is comparatively close. However, as the dimension *d* increases, the approximate mixing time of PLMC and MYULA increases rapidly, showing a polynomial order with respect to *d*. Moreover, the dimension dependence of MLD and Algorithm 2 both indicate a linear growth trend, and MLD needs a few more steps than Algorithm 2 to reach the same error tolerance.

Figure 4b presents the dependence of the approximate mixing time ${\hat{k}}_{\mathrm{mix}}\left(\varepsilon \right)$ on the inverse of the error tolerance $\varepsilon^{-1}$ for the four sampling algorithms under d=4. The regression analysis shows that the approximate mixing time ${\hat{k}}_{\mathrm{mix}}\left(\varepsilon \right)$ of PLMC and MYULA increases in polynomial order of $\varepsilon^{-1}$. When $\varepsilon^{-1}$ is relatively small, MLD and Algorithm 2 have similar approximate mixing time. With the increase in $\varepsilon^{-1}$, the strength of Algorithm 2 gets more significant. For MLD, the linear regression for the approximate mixing time ${\hat{k}}_{\mathrm{mix}}\left(\varepsilon \right)$ with respect to $\varepsilon^{-2}$ yields a slope of 1.934 and *R*-squared 0.984, suggesting the error tolerance dependence of order $\varepsilon^{-2}$.

It is noteworthy that the above analysis not only suggests significantly better dimension and error tolerance dependence of the constrained MALA but also partly verifies the theoretical convergence rates of the three methods for comparison.

### 5.2. Bayesian Regularized Regression

The regularized regression involves adding a penalty term on the objective function of the regression model, which helps to control the complexity of the model and prevent it from fitting the noise in the data. In this section, we validate the effectiveness of Algorithm 3 for constrained sampling involving the Bayesian regularized regression.

Given the independent and identically observations $y={\left(y_{1},y_{2},\ldots ,y_{n}\right)}^{T}\in R^{n}$ which follow from the Gaussian distribution with mean Xβ and covariance matrix $\sigma^{2}I_{n}$, we consider the regression models where the parameter are obtain by minimizing the square of Euclidean norm of the residual subject to a norm-constraint on the regression parameter as follows:$$\min_{\beta \in R^{d}}{\left|y-X\beta \right|}_{2}^{2}\;\;\;\mathrm{subject}\;\mathrm{to}\;\;{\left|\beta \right|}_{p}\leq C$$
for some universal constant C>0, where $X\in R^{n\times d}$ is the design matrix, $\beta \in R^{d}$ is the regression parameter, and ${\left|\beta \right|}_{p}$ is the $L_{p}$-norm of β. In Bayesian setting, many regularization techniques correspond to imposing certain prior distributions on model parameters. We then consider sampling from the distribution with density
$$\pi^{*}\left(x\right)\propto \exp\left\{-\frac{{\left|y-X\beta \right|}_{2}^{2}}{2\sigma^{2}}\right\}I\left(x\in X\right)\;,$$
and obtaining the parameter estimates $\hat{\beta }$ via the maximum a posteriori probability (MAP) estimate, where $X=\{x\in R^{d}:|x{|}_{p}\leq C\}$. We use the diabetes data studied in [41], and set the burn-in period to be $10^{3}$ iterations and $\sigma^{2}=1$. Figure 5 presents the paths of the parameter estimates under different norm constraints, which demonstrate that Algorithm 3 can effectively handle the norm-constrained sampling problems.

### 5.3. Truncated Multivariate Gaussian Distribution

The final comparison was made by examining the sampling performance of MYULA in [35] and Algorithm 4 in the setting of a more general truncated multivariate Gaussian distribution. We consider the same setup as in [35]. Specifically, the density of the target distribution is defined as follows:$$\pi^{*}\left(x\right)\propto \exp\left\{-\frac{{\left(x-\mu \right)}^{T}\Sigma^{-1}\left(x-\mu \right)}{2}\right\}I\left(x\in X\right)\;,$$
where X is a convex set and the origin 0 is on its boundary. Let μ=0, the covariance matrix $\Sigma \in R^{d\times d}$ with (i,j)-th element given by ${\left(\Sigma \right)}_{i,j}=1/\left(1+|i-j|\right)$, and X=[0,5]×[0,1]. We generate $10^{6}$ samples for Algorithm 4, and set the burn-in period to be the initial 10% iterations.

Table 3 presents the mean and covariance estimation results of the target distribution based on the samples generated by MYULA and Algorithm 4. For comparison purposes, the results of MYULA align with those reported in [35]. With the same number of iterations, Algorithm 4 outperforms MYULA in terms of the estimation results. This indicates that incorporating the Metropolis–Hastings step in Algorithm 4 leads to improvements in the mixing time.

## 6. Discussion and Conclusions

In this article, we propose three sampling algorithms based on Langevin Monte Carlo with the Metropolis–Hastings steps to handle the distribution constrained within some convex body, and establish the mixing time bounds of these algorithms for sampling from strongly log-concave distributions. Under certain conditions, these bounds are sharper than existing algorithms in the literature. Furthermore, in comparison to existing algorithms, the suggested constrained sampling algorithms are simpler, more intuitive, and easier to operate in some cases.

Our results demonstrate that the sampling algorithm, enhanced with the Metropolis–Hastings step, offers an effective solution for tackling some constrained sampling problems. Numerical experiments fully illustrate the advantages of the proposed algorithms. Although we focus on the strongly log-concave distributions in the theoretical analysis, the proposed algorithm can be readily applied to weakly log-concave distributions or non-convex potential functions. Simultaneously, we recognize that there are various aspects of the sampling algorithms that can be further improved. For instance, potential enhancements could involve the multiple importance sampling methods or adaptive techniques. We leave the investigation of its theoretical properties under such scenarios for future work.

## Figures and Tables

**Figure 1 entropy-25-01234-f001:**
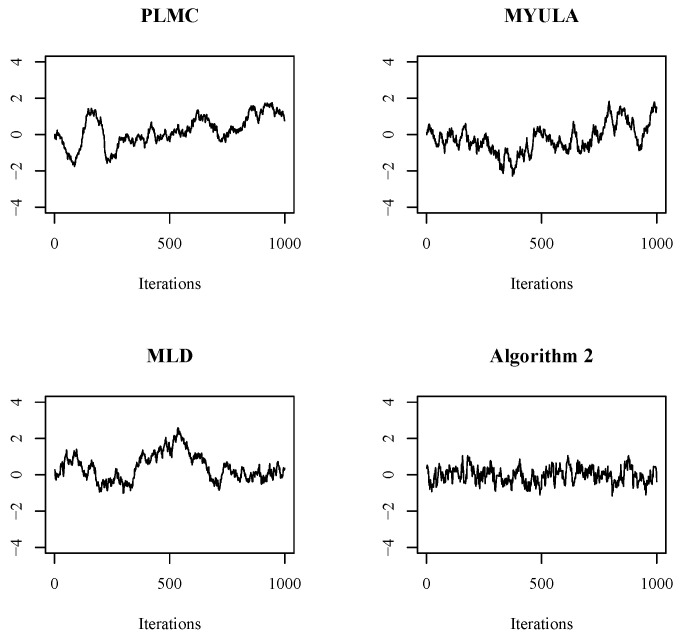
The trace graphs of $x_{1}$ of the Markov chain determined by the four sampling algorithms.

**Figure 2 entropy-25-01234-f002:**
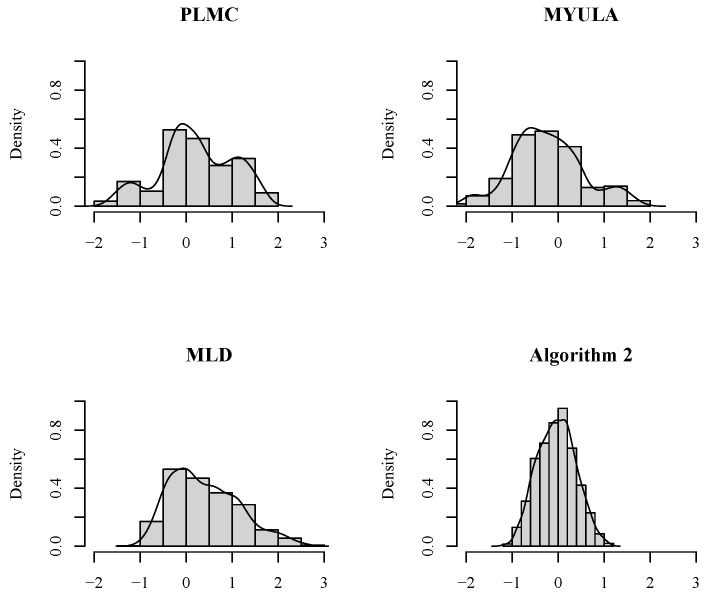
The densities of $x_{1}$ of the Markov chain determined by the four sampling algorithms.

**Figure 3 entropy-25-01234-f003:**
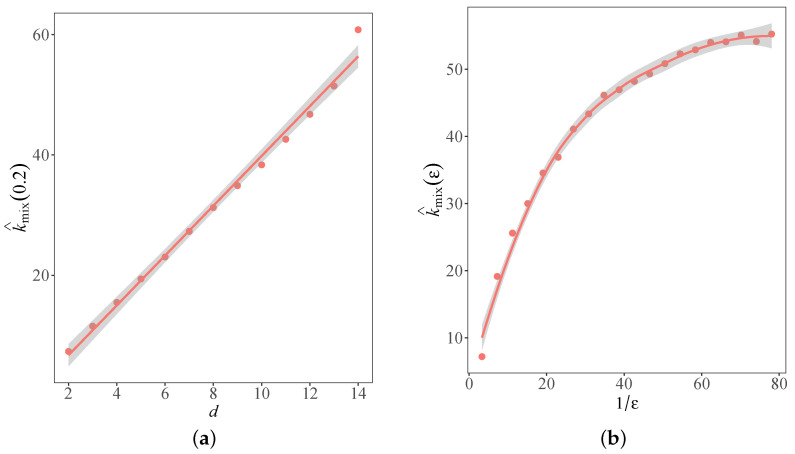
Approximate mixing time with respect to dimension and error tolerance of Algorithm 2. (**a**) Dimension dependence for fixed error tolerance. (**b**) Error tolerance dependence for fixed dimension.

**Figure 4 entropy-25-01234-f004:**
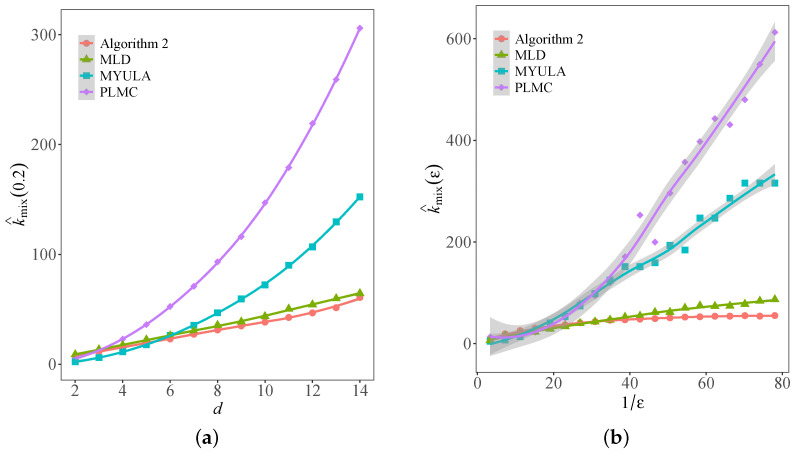
Approximate mixing time with respect to dimension and error tolerance dependence of the four sampling algorithms. (**a**) Dimension dependence for fixed error tolerance. (**b**) Error tolerance dependence for fixed dimension.

**Figure 5 entropy-25-01234-f005:**
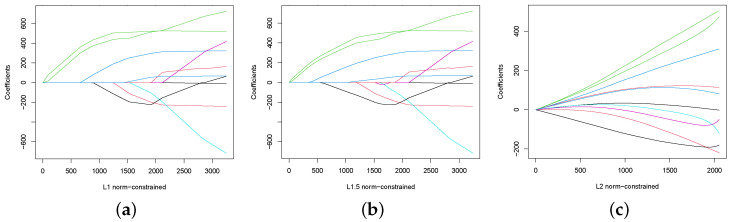
Bayesian regularized regression via Algorithm 3, where distinct colors represent various trajectories of parameter estimates for distinct variables. (**a**) $L_{1}$—norm-constraint. (**b**) $L_{1.5}$—norm-constraint. (**c**) $L_{2}$—norm-constraint.

**Table 1 entropy-25-01234-t001:** Convergence rates for sampling from log-concave distributions with bounded support.

Assumptions	${∥\cdot ∥}_{\mathrm{TV}}$ Rate	Algorithms
$0I_{d}⪯\nabla^{2}U\left(x\right)⪯LI_{d}$	$\tilde{O}\left(d^{12}\varepsilon^{-12}\right)$	PLMC in [34]
$mI_{d}⪯\nabla^{2}U\left(x\right)⪯LI_{d}$	$\tilde{O}\left(d^{5}\varepsilon^{-6}\right)$	MYULA in [35]
$mI_{d}⪯\nabla^{2}U\left(x\right)$	$\tilde{O}\left(d\varepsilon^{-2}\right)$	MLD in [36]
$mI_{d}⪯\nabla^{2}U\left(x\right)⪯LI_{d}$	$\tilde{O}\left\{d\log\left(1/\varepsilon \right)\right\}$	Algorithms 2 and 3 in this paper
$mI_{d}⪯\nabla^{2}U\left(x\right)⪯LI_{d}$	$\tilde{O}\left(d^{3}\varepsilon^{-2}\right)$	Algorithm 4 in this paper

**Table 2 entropy-25-01234-t002:** Step sizes for sampling from log-concave distributions with bounded support.

Algorithms	Step Size
PLMC in [34]	$L^{-1}d^{-2}$
MYULA in [35]	${\left\{d\max\left(d,L\right)\right\}}^{-1}$
MLD in [36]	the grid search
Algorithm 2 in this paper	$L^{-7/4}R^{-3/2}d^{-1}$
Algorithm 3 in this paper	$L^{-7/4}d^{-1}$
Algorithm 4 in this paper	${\left\{\left(L+\lambda^{★-1}\right)\max\left[d,{\left\{m^{-1}d\left(L+\lambda^{★-1}\right)\right\}}^{1/2}\right]\right\}}^{-1}$

**Table 3 entropy-25-01234-t003:** The mean and covariance estimation results obtained by MYULA and Algorithm 4.

Assumptions	Mean	Covariance
The truth	$\left(\begin{array}{c} 0.790\\ 0.488 \end{array}\right)$	$\left(\begin{array}{cc} 0.326 & 0.017\\ 0.017 & 0.080 \end{array}\right)$
MYULA	$\left(\begin{array}{c} 0.758\pm 0.052\\ 0.484\pm 0.016 \end{array}\right)$	$\left(\begin{array}{cc} 0.309\pm 0.038 & 0.017\pm 0.009\\ 0.017\pm 0.009 & 0.088\pm 0.002 \end{array}\right)$
Algorithm 4	$\left(\begin{array}{c} 0.781\pm 0.034\\ 0.491\pm 0.009 \end{array}\right)$	$\left(\begin{array}{cc} 0.317\pm 0.012 & 0.017\pm 0.004\\ 0.017\pm 0.004 & 0.082\pm 0.003 \end{array}\right)$

## Data Availability

The data used to support the findings of this study are included within the article.

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
