# Peer review of "Convergence Rates for the Constrained Sampling via Langevin Monte Carlo"

_entropy, 2023, doi:10.3390/e25081234_

Round 1

Reviewer 1 Report

The authors propose three sampling algorithms based on Langevin
Monte Carlo with the Metropolis-Hastings steps to handle the distribution constrained within some convex body. Their results demonstrate that the sampling algorithm, enhanced with the Metropolis-Hastings step, offers an
effective solution for tackling some constrained sampling problems. The numerical experiments are conducted to compare our methods with several competing algorithms without Metropolis-Hastings
steps, and the results further support our theoretical findings.

In my opinion, this is a good manuscript for publication.

Metropolis-Hastings step--> the Metropolis-Hastings step
machine learning -->machine-learning
More recently, [30,31] have-->More recently, [30,31] has
 case, and derive--> case and derived
for more -->for the more
algorithms can also leads -->algorithms can also lead
the Appendix B.--> Appendix B.
certain metric--> certain metrics
a ε-accurate-->an ε-accurate

Author Response

Thank you for your valuable feedback. I have carefully corrected the grammatical, styling, and typos found in the manuscript.

Reviewer 2 Report

The authors propose three sampling algorithms based on Langevin Monte Carlo with Metropolis-Hastings steps to handle the distribution constrained within some covex body in this paper. The total variation distance is used to prove a rigorous analysis of Markov chains and non-asymptotic upper bounds on the convergence rates of these algorithms. Extensive numerical experiments demonstrate the effectiveness of the proposed three sampling algorithms. This paper is well written and includes both theoretical justification and numerical experiments. A minor revision should carefully address the following comments.

1: The authors assume that X is a compact and convex set and that U is an m-strongly convex function in Assumption 1 and 2. Could the authors provide comments on additional cases, such as non-convex set X and non-convex function U?

2:  The density $p^0$ of the initial distribution $\mathbb{P}^{0}$ is quite close to $\pi(x)$. Could algorithms also work well for $p^0$, which is far away from both prior and posterior density.

3: The theoretical properties of three algorithms are validated by interesting and rigorous numerical examples. Please show some plots of Markov chains or sample histograms if possible.

Author Response

Thank you for your insightful comments and valuable feedback.

Reviewer 3 Report

An improvement is required.

Author Response

(The authors gave the same response as above.)

Round 2

Reviewer 2 Report

The authors answered all my questions very carefully and addressed all issues appropriately with new numerical results. I recommend publishing this work.

Reviewer 3 Report

The reviewer thanks the authors for considering the comments carefully. The paper is recommended for publication.

fine.